# Association between serum ferritin and mortality in patients with severe fever with thrombocytopenia syndrome: A retrospective cohort study

Wenyan Xiao[1,2◉], Liangliang Zhang[1,2◉], Yang Zhang[1,2◉], Juanjuan Hu[1,2], Jin Zhang[1,2], Tianfeng Hua[1,2‡]*, Min Yang [1,2‡]*

1 The Second Department of Critical Care Medicine, the Second Affiliated Hospital of Anhui Medical University, Anhui, Hefei, P.R.China, 2 The Laboratory of Cardiopulmonary Resuscitation and Critical Care Medicine, the Second Affiliated Hospital of Anhui Medical University, Anhui, Hefei, P.R.China

◉ These authors contributed equally to this manuscript.
‡ These corresponding authors contributed equally to this manuscript.
* huatianfeng@ahmu.edu.cn (TH); yangmin@ahmu.edu.cn (MY)

## Abstract

The role of serum ferritin, an acute-phase inflammatory marker, in predicting mortality in patients with severe fever with thrombocytopenia syndrome (SFTS) is not fully understood. This study aimed to investigate the association between serum ferritin levels and inpatient mortality in patients with SFTS. We conducted a retrospective analysis using data from patients diagnosed with SFTS at the Second Affiliated Hospital of Anhui Medical University between May 2017 and September 2024. The association between serum ferritin levels and in-hospital mortality in patients with SFTS was assessed using Cox regression models and restricted cubic spline (RCS) analysis. The receiver operating characteristic (ROC) curve was used to determine the optimal serum ferritin cut-off value for predicting mortality risk. Kaplan–Meier survival curves were compared for survival rates, and propensity score matching with linear trend testing was applied to ensure robustness. This study included 390 patients with SFTS, of whom 312 survived and 78 did not, yielding an in-hospital mortality rate of 20.0%. Cox regression and RCS analyses demonstrated a significant linear association between higher serum ferritin levels and increased in-hospital mortality in patients with SFTS, indicating that the predicted mortality of patients with SFTS increased with elevated serum ferritin levels beyond a certain threshold. ROC analysis revealed an area under the curve of 0.830, with an optimal serum ferritin cut-off of 3.975 (lg, ng/ml), sensitivity of 0.731, and specificity of 0.830. Clinically, serum ferritin levels above 4 (lg ng/ml), were associated with a substantial increase in mortality risk. Sensitivity analysis supported the robustness of these results. Serum ferritin levels are linearly associated with mortality risk in patients with SFTS, with mortality significantly increasing when serum ferritin levels exceed 10,000 ng/ml.

**Data availability statement:** All relevant data are within the manuscript and it's Supporting Information files.

**Funding:** The study was supported by National Natural Science Foundation of China (grant number 82072134 to YM), the Research Fund of Anhui Institute of translational medicine (grant number 2022zhyx-C46 to XWY), the Health Research Program of Anhui (grant number AHWJ2022b085 to XWY) and the Natural Science Research Project Funding of Higher Education Institutions Anhui Province (grant number 2023AH040375 to XWY). The funders had no role in the study design, data collection and analysis, decision to publish, or preparation of the manuscript.

**Competing interests:** The authors declare that they have no competing interests.

Serum ferritin level may serve as a valuable prognostic biomarker for mortality risk in patients with SFTS.

## Author summary

Fever with thrombocytopenia syndrome (SFTS) is an acute infectious disease caused by SFTS virus, and 12%-30% of patients die of multi-organ failure, in which the extreme inflammatory response induced by SFTS virus plays a key role in the pathogenesis. Previous studies on the clinical prognosis of SFTS have focused on inflammatory indicators such as C-reactive protein and interleukin-6, but their specificity and sensitivity are poor. Serum ferritin, as an acute inflammatory protein, has been shown to be associated with poor prognosis of viral infectious diseases such as COVID-19, influenza, and dengue fever, but the relationship with the prognosis of SFTS is unclear. Here, we systematically evaluated the relationship between serum ferritin and in-hospital mortality in patients with SFTS by constructing different proportional risk models. Our results showed a significant linear relationship between serum ferritin on in-hospital mortality in SFTS patients, and serum ferritin showed good specificity and sensitivity in predicting death in SFTS, especially, when serum ferritin levels exceed 10,000 ng/ml, the in-hospital mortality rate of SFTS patients was significantly higher. The above results were further validated by Propensity score matching and linear trend test.

## Background

Severe fever with thrombocytopenia syndrome (SFTS) is an acute infectious disease caused by the Dabie-Banzo virus (DBV), also known as SFTS virus (SFTSV). First reported in 2009, this disease has spread rapidly across multiple countries, including China, Japan, South Korea, the United States, and the United Arab Emirates [1]. Patients with SFTS typically present with a range of symptoms including acute fever, thrombocytopenia, gastrointestinal issues, and central nervous system complications [2]. However, in some cases, the disease progresses rapidly to multiorgan failure, resulting in high mortality rates ranging from 12–30% [3–4].

Serum ferritin is a multifunctional protein that serves as a marker of iron storage and as an acute-phase reactant in inflammatory processes. Serum ferritin levels fluctuate in response to various disease states, including infections, inflammatory conditions, and cancers [5]. Serum ferritin, released from macrophages during acute infection or inflammation, has been identified as a key marker of macrophage activation, and its elevated levels reflect the intensity of the immune response to infection and inflammation [6]. Several studies have linked elevated serum ferritin levels to poor prognosis in various viral infections, including COVID-19, influenza, and dengue [7–9].

Specifically, Weng et al. reported elevated serum ferritin levels in patients with SFTS, with some patients experiencing severe inflammatory responses, such as hemophagocytic lymphohistiocytosis (HLH) [10]. These findings suggest that SFTSV infection may trigger an excessive immune response, leading to a significant increase in inflammatory biomarkers such as serum ferritin. However, the relationship between serum ferritin levels and mortality in patients with SFTS remains unclear. Therefore, this retrospective cohort study investigated the impact of serum ferritin levels on the prognosis of patients with SFTS, aiming to encourage early assessment and timely clinical intervention for patients with SFTS with elevated ferritin levels.

## Methods

### Ethics approval and consent to participate

The Ethics Committee of the Second Affiliated Hospital of Anhui Medical University approved the study and waived the requirement for informed consent, given the retrospective nature of the study (No. YX2022–041).

### Data sources and study population

This study utilized data from demographic characteristics, clinical manifestations, and laboratory test results of patients diagnosed with SFTS at the Second Affiliated Hospital of Anhui Medical University between May 2017 and September 2024.

Diagnostic criteria: SFTS was diagnosed according to the 2010 guidelines for the prevention and treatment of SFTS [11]. Diagnostic criteria included both suspected cases and serological evidence. Suspected cases involved patients with an epidemiological history such as history of working, living or traveling in hilly areas during the epidemic season, or history of tick bite within two weeks prior to the onset of the disease, body temperature ≥38 °C on admission, symptoms such as fatigue and nausea, muscle pains, and blood tests showing reduced leukocytes and platelets at admission. Serological evidence required one of the following: (1) positive SFTSV nucleic acid detection, (2) an increase in IgG antibody titer during recovery of >4 times compared to the acute phase, (3) isolation of SFTSV, or (4) detection of SFTS-specific IgM antibodies in acute phase serum. Exclusion criteria included: (1) severe clinical data deficiency; (2) immunodeficiency, malignancy, hematological diseases; and (3) clinically confirmed rickettsial infections, hemorrhagic fever with renal syndrome, dengue fever, and other infectious diseases.

### Variables and data processing

The variables included (1) demographic information (age, sex, and occupation); (2) comorbidities (e.g., diabetes, hypertension, and stroke); (3) symptoms (e.g., diarrhea, neurological symptoms, gastrointestinal bleeding, pulmonary fungal infections, and bacteremia; (4) vital signs at admission (heart rate, respiratory rate, systolic and diastolic blood pressure; and (5) laboratory tests, including SFTSV RNA levels, blood counts (e.g., white blood cells, neutrophils, lymphocytes, hemoglobin, and platelets), liver function (e.g., alachlor aminotransferase, alanine aminotransferase, and total bilirubin), renal function (urea nitrogen, creatinine), cardiac enzymes (lactate dehydrogenase, creatine kinase, creatine kinase MB), coagulation markers (plasma plasminogen time, prothrombin time, fibrinogen, D-dimer), and inflammatory markers (C-reactive protein, procalcitonin, and serum ferritin). All laboratory data were collected within 24 h of patient admission, using only the initial test results when multiple tests were conducted within this period.

SFTSV RNA detection was performed using real-time quantitative reverse transcription-polymerase chain reaction (Applied Biosystems 7500, USA), which has a lower detection threshold of 1000 copies/mL and a negative record of 1000 copies/mL. Twenty-three patients with SFTS had a negative test for SFTSV nucleic acid and positive IgM-specific antibody, and the nucleic acid record of this subset of patients was 1000 copies/mL. SFTSV RNA and serum ferritin levels were determined by log10 conversion.

Definitions for specific conditions were as follows: gastrointestinal hemorrhage was defined as the presence of blood in stool or vomit, black stool, or a positive fecal or vomit occult blood test (2+ or higher). Pancreatitis was defined by epigastric pain, amylase or lipase levels three times above normal, and abdominal CT findings consistent with pancreatic exudative changes in two of three ways; fungal infection in the lungs was diagnosed according to clinical criteria in the relevant guidelines [12]. Neurological symptoms included impaired consciousness, convulsions, or delirium. Bacteremia was defined by pathogen identification in blood cultures and exclusion in contaminated blood specimens.

Serum ferritin levels served as the primary study variable with in-hospital mortality as the primary outcome and length of stay as a secondary outcome.

### Statistical analysis

The covariance between variables was checked using the variance inflation factor (VIF), with VIF values >10 leading to exclusion of the variable. Variables with >10% missing values were excluded, whereas missing data for others were imputed using the random forest multiple imputation method. Variables that conformed to a positive distribution are described by mean±standard deviation, and differences between the two groups were analyzed using independent sample $t$-tests. Continuous variables that did not conform to a positive distribution were reported as medians with interquartile ranges and analyzed using Mann–Whitney U tests. Categorical variables were reported as frequencies and percentages, and differences were analyzed using the chi-square or Fisher's exact tests.

Four Cox proportional risk models were constructed to assess the correlation between serum ferritin level and in-hospital mortality: crude model (unadjusted), Model I (adjusted for age, sex, and occupation), Model II (adjusted for age, sex, occupation, and underlying disease), and Model III (adjusted for all covariates).

RCS curves were used to explore the relationship between serum ferritin levels and in-hospital mortality across these models. ROC analysis was used to determine the optimal ferritin cut-off for mortality risk, and patients were divided into high- and low-risk groups. Survival curves were plotted using the Kaplan–Meier method to compare the survival rates of the two groups.

Sensitivity analyses were conducted using two approaches: PSM was performed to minimize confounding bias for age, SFTSV RNA, platelet count, C-reactive protein, and Pcalcitoninogen, with a 1:1 match and a caliper width of 0.05, and serum ferritin was transformed into a categorical variable with trend tests conducted via Cox regression.

All statistical analyses were performed using R version 4.2.3, and statistical significance was set at $P < 0.05$.

## Results

### Baseline characteristics of patients with SFTS

The data of 462 patients with confirmed SFTS were obtained, and 72 patients were excluded, resulting in 390 patients included in the study (Fig 1). Of these, 161 (41.3%) were male and 229 (58.7%) were female, and 78 (20.0%) experienced in-hospital mortality. Patients in the mortality group had a significantly higher median age than those in the survival group (73 vs. 67 years, $P < 0.001$). Patients in the mortality group were more likely to develop complications such as pancreatitis, pulmonary fungal infections, gastrointestinal hemorrhage, and bacteremia ($P < 0.001$). Viral load, Pcalcitoninogen, C-reactive protein, and serum ferritin levels were significantly elevated in the mortality group ($P < 0.001$). Platelet and white blood cell counts were lower in patients in the mortality group than in the survival group (p<0.001) (Table 1).

### Association of serum ferritin with in-hospital mortality in SFTS in different Cox proportional risk models

Cox regression analyses were performed to assess the relationship between serum ferritin and in-hospital mortality in patients with SFTS across four models. The hazard ratio (HR, 95% CI) for serum ferritin in the Crude model was 5.950

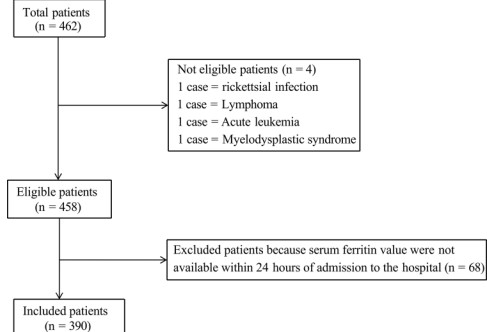

**Fig 1. Study fow chart.** Flow diagram for patient selection and cohort conformation.

(4.059–8.720) (S1 Table), whereas in Model I, it was 5.968 (3.969–8.975) (S2 Table). In Model II, the HR was 5.714 (3.780–8.639) (S3 Table), and in Model III, the HR was 5.982 (2.752–13.004) (S4 Table). These results underscore the diagnostic value of serum ferritin as a predictor of prognosis in patients with SFTS under different conditions ($P<0.001$) (Table 2).

Using Cox regression-based restricted cubic spline (RCS) curves, the relationship between serum ferritin levels and in-hospital mortality was further evaluated using the four models, and a significant linear relationship was observed ($P$ for nonlinearity $>0.05$) (Fig 2).

### Receiver operating characteristic (ROC) curve of serum ferritin for predicting in-hospital mortality in patients with SFTS

The ROC curve for serum ferritin showed an area under the curve of 0.830, with an optimal cut-off value of 3.975 (lg, ng/ml), sensitivity of 0.731, and specificity of 0.830 (Fig 3), indicating strong predictive value.

### Survival curves of patients in the high and low serum ferritin groups

For clinical relevance, the best cut-off value of serum ferritin was rounded off and divided into two groups, with levels ≥ 4 (log, ng/mL) defining the high serum ferritin group and levels < 4 (log, ng/ml) defining the low serum ferritin group. The Kaplan–Meier survival analysis showed that patients in the high serum ferritin group (≥10,000 ng/ml) had significantly lower survival than patients in the low serum ferritin group (<10,000 ng/ml), with a statistically significant difference between the two groups ($P<0.001$) (Fig 4).

### Propensity score matching and linear trend test

To further explore the correlation between serum ferritin levels and in-hospital mortality in patients with SFTS, propensity score matching (PSM) was used to reduce the confounding bias of age, SFTSV RNA, platelet count, C-reactive protein, and procalcitonin levels and to compare the differences in serum ferritin between survivors and non-survivors after matching (S1 Fig). A total of 102 patients with SFTS had higher serum ferritin levels in the mortality group after PSM [(4.15±0.52) vs. (3.86±0.52), $P=0.007$] (Table 3).

Serum ferritin was analyzed as a categorical variable using quartiles, with Cox regression performed under each of the four risk models. A trend test was also conducted (Table 4). Although the Q2 and Q3 quartiles did not show significant risk elevation compared to the Q1 group, the Q4 group showed a notable increase in mortality risk. The trend test confirmed a significant positive association between serum ferritin levels and increased mortality risk. ($P$ for trend $<0.01$).

**Table 1. Baseline characteristics of SFTS patients.**

| Variables | Total (n = 390) | Survivor (n = 312) | Nonsurvivor (n = 78) | P |
|---|---|---|---|---|
| **Demographic feature** | | | | |
| Gender, Male (%) | 161 (41.3) | 122 (39.1) | 39 (50.0) | 0.080 |
| Age, years | 68 (58, 74) | 67 (58, 73) | 73 (67, 79) | < 0.001 |
| Farmer, n (%) | 334 (85.6) | 270 (86.5) | 64 (82.1) | 0.312 |
| **Symptoms** | | | | |
| Diarrhea, n (%) | 177 (45.4) | 136 (43.6) | 41 (52.6) | 0.154 |
| Pancreatitis, n (%) | 207 (53.1) | 145 (46.5) | 62 (79.5) | < 0.001 |
| Neurologic symptoms, n (%) | 111 (28.5) | 36 (11.5) | 75 (96.2) | < 0.001 |
| Gastrointestinal bleeding, n (%) | 82 (21.0) | 37 (11.9) | 45 (57.7) | < 0.001 |
| Pulmonary fungal infection, n (%) | 98 (25.1) | 47 (15.1) | 51 (65.4) | < 0.001 |
| Bacteremia, n (%) | 27 (6.9) | 15 (4.8) | 12 (15.4) | < 0.001 |
| **Comorbidity** | | | | |
| Hypertension, n (%) | 119 (30.5) | 82 (26.3) | 37 (47.4) | < 0.001 |
| Diabetes, n (%) | 36 (9.2) | 28 (9.0) | 8 (10.3) | 0.726 |
| Stroke, n (%) | 35 (9.0) | 21 (6.7) | 14 (17.9) | 0.002 |
| **Vital signs** | | | | |
| Temperature, (°C) | 37.0 (36.6, 38.0) | 37.0 (36.5, 38.0) | 37.3 (36.7, 38.3) | 0.029 |
| Respiratory rate, (min$^{-1}$) | 19 (18, 20) | 19 (18, 20) | 20 (18, 20) | 0.012 |
| Heart rate, (min$^{-1}$) | 79 (70, 89) | 78 (69, 87) | 86 (74, 98) | < 0.001 |
| Systolic BP, (mmHg) | 115 ± 18 | 114 ± 18 | 120 ± 19 | 0.011 |
| Diastolic BP, (mmHg) | 69 (62, 76) | 69 (62, 76) | 70 (62, 79) | 0.157 |
| **Laboratory tests** | | | | |
| SFTSV RNA, (lg, copies/ml) | 5.77 (4.32, 6.56) | 5.38 (4.20, 6.20) | 6.91 (6.23, 7.79) | < 0.001 |
| White blood cell, (K/uL) | 2.06 (1.43, 3.23) | 2.06 (1.44, 3.22) | 2.14 (1.37, 3.35) | 0.750 |
| Hemoglobin, (g/L) | 128 (120, 141) | 128 (119, 141) | 128 (121, 141) | 0.829 |
| Platelet, (K/uL) | 52 (37, 72) | 55 (40, 74) | 41 (30, 54) | < 0.001 |
| Glutamic oxaloacetic transaminase, (U/L) | 146 (74, 334) | 126 (68, 252) | 302 (153, 587) | < 0.001 |
| Total bilirubin, (µmol/L) | 9.3 (7.3, 11.9) | 9.1 (7.1, 11.5) | 9.6 (8.0, 13.3) | 0.045 |
| Creatinine, (µmol/L) | 74 (59, 99) | 70 (56, 88) | 112 (80, 144) | < 0.001 |
| Urea nitrogen, (mmol/L) | 6.60 (5.19, 9.43) | 6.35 (4.91, 8.40) | 10.13 (6.66, 15.44) | < 0.001 |
| Lactic dehydrogenase, (U/L) | 518 (328, 886) | 439 (311, 727) | 897 (557, 1632) | < 0.001 |
| Creatine kinase, (U/L) | 317 (137, 839) | 267 (127, 705) | 607 (223, 1485) | < 0.001 |
| Creatine kinase MB, (U/L) | 29 (19, 46) | 27 (18, 42) | 39 (27, 66) | < 0.001 |
| Plasma prothrombin time, (s) | 11.5 (10.8, 12.3) | 11.4 (10.7, 12.2) | 12.0 (11.2, 13.1) | < 0.001 |
| D-D dimer, (µg/ml) | 2.57 (1.36, 6.16) | 2.12 (1.21, 4.97) | 6.64 (2.81, 13.63) | < 0.001 |
| Thrombin time, (s) | 22.0 (19.5, 27.7) | 21.1 (19.3, 24.9) | 29.7 (22.9, 46.1) | < 0.001 |
| Fibrinogen, (g/L) | 2.32 (1.97, 2.68) | 2.38 (1.99, 2.17) | 2.11 (1.84, 2.52) | 0.002 |
| C-reactive protein, (mg/L) | 2.8 (0.7, 7.8) | 2.6 (0.6, 5.9) | 6.1 (2.3, 17.4) | < 0.001 |
| Procalcitonin, (ng/ml) | 0.160 (0.080, 0.340) | 0.136 (0.070, 0.270) | 0.455 (0.179, 2.075) | <0.001 |
| Serum ferritin, (lg, ng/ml) | 3.62 ± 0.61 | 3.48 ± 0.56 | 4.18 ± 0.48 | < 0.001 |
| Hospital stay, (days) | 12 (8, 16) | 13 (11, 17) | 5 (3, 6) | < 0.001 |

**Table 2. Hazard ratios of serum ferritin for the four cox proportional hazard models.**

| Model | HR, 95%CI | P |
|---|---|---|
| Crude model | 5.950 (4.059-8.720) | <0.001 |
| Model I [a] | 5.968 (3.969-8.975) | <0.001 |
| Model II [b] | 5.714 (3.780-8.639) | <0.001 |
| Model III [c] | 5.982 (2.752-13.004) | <0.001 |

[a]Model I: Adjusted for age, sex, and profession.

[b]Model II: Adjusted for age, sex, profession, and comorbidity.

[c]Model III: Adjusted for all covariates.

Abbreviations: HR, hazard ratio; CI, confidence interval

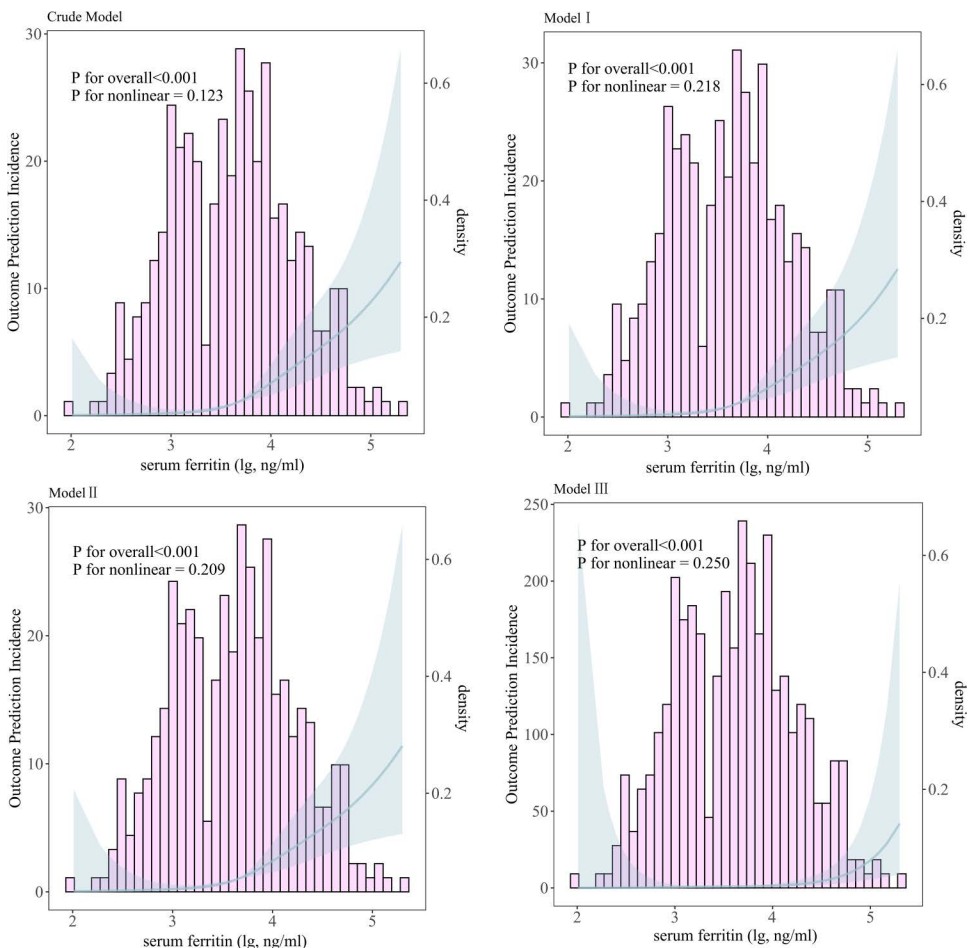

**Fig 2. The relationship between serum ferritin and in-hospital mortality under the four risk models using Cox regression-based RCS curves.**
Model I: Adjusted for age, sex, and profession. Model II: Adjusted for age, sex, profession, and comorbidity. Model III: Adjusted for all covariates.

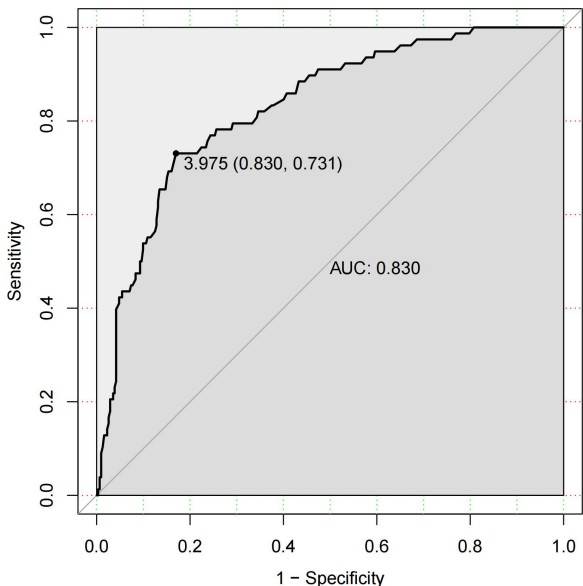

**Fig 3. ROC curve of serum ferritin for predicting in-hospital mortality.**

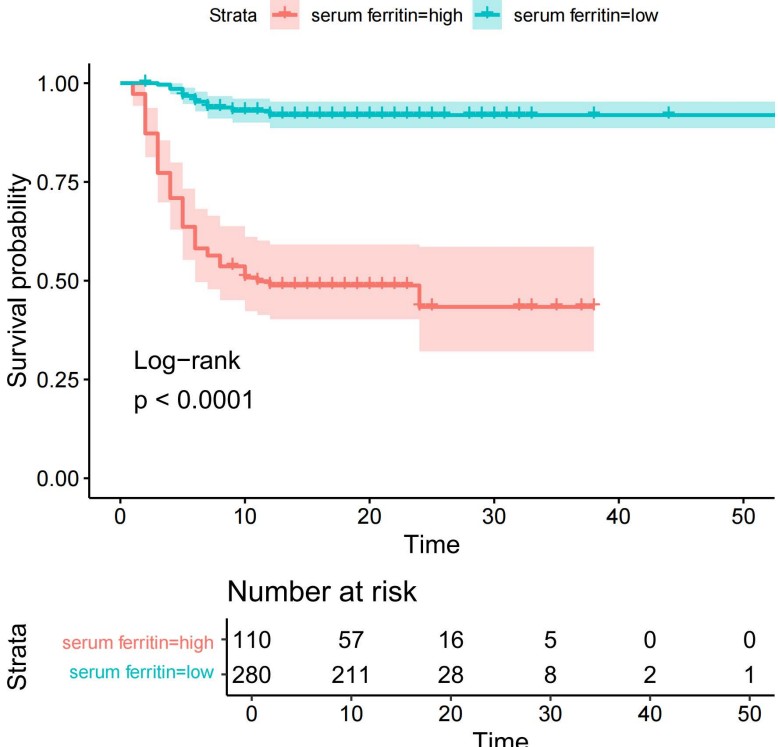

**Fig 4. Kaplan-Meier survival analysis comparing patients categorized into high and low serum ferritin groups. high serum ferritin group ( ≥ 10,000 ng/ml); low ferritin group (< 10,000 ng/ml).**

**Table 3. Clinical comparison of SFTS patients with propensity score matched analysis.**

| Variables | Before PSM | | | | After PSM | | | |
|---|---|---|---|---|---|---|---|---|
| | Survivor (n = 312) | Nonsurvivor (n = 78) | SMD | P | Survivor (n = 51) | Nonsurvivor (n = 51) | SMD | P |
| Age | 67 (58, 73) | 73 (67, 79) | 0.777 | < 0.001 | 73 (68, 76) | 73 (66, 80) | 0.097 | 0.799 |
| SFTSV RNA | 5.38 (4.20, 6.20) | 6.91 (6.23, 7.79) | 1.521 | < 0.001 | 6.60 ± 0.84 | 6.52 ± 0.85 | -0.032 | 0.647 |
| PLT | 55 (40, 74) | 41 (30, 54) | -0.552 | < 0.001 | 40 (33, 61) | 40 (32, 60) | 0.086 | 0.941 |
| CRP | 2.6 (0.6, 5.9) | 6.1 (2.3, 17.4) | 0.300 | < 0.001 | 4.7 (1.3, 11.8) | 5.7 (2.3, 13.8) | -0.004 | 0.437 |
| PCT | 0.136 (0.070, 0.270) | 0.455 (0.179, 2.075) | 0.441 | <0.001 | 0.260 (0.130, 0.590) | 0.405 (0.131, 1.108) | -0.023 | 0.092 |
| Serum ferritin | 3.48 ± 0.56 | 4.18 ± 0.48 | | < 0.001 | 3.86 ± 0.52 | 4.15 ± 0.52 | | 0.007 |

PSM, propensity score matched; CRP, C-reactive protein; PCT, Procalcitonin.

**Table 4. Linear trend test of serum ferritin in four cox proportional hazard models.**

| Serum ferritin Quartile | Crude model | | Model I [a] | | Model II [b] | | Model III [c] | |
|---|---|---|---|---|---|---|---|---|
| | HR, 95%CI | P | HR, 95%CI | P | HR, 95%CI | P | HR, 95%CI | P |
| Q1 | Ref | | Ref | | Ref | | Ref | |
| Q2 | 4.487 (0.969-20.770) | 0.055 | 4.251 (0.916-19.729) | 0.065 | 4.155 (0.894-19.314) | 0.069 | 5.563 (0.777-39.822) | 0.087 |
| Q3 | 8.000 (1.839-34.800) | 0.006 | 7.001 (1.608-30.487) | 0.010 | 6.924 (1.587-30.218) | 0.010 | 1.900 (0.289-12.506) | 0.505 |
| Q4 | 34.533 (8.403-141.920) | <0.001 | 30.036 (7.294-123.684) | <0.001 | 28.820 (6.967-119.224) | <0.001 | 10.780 (1.578-73.652) | 0.015 |
| P for trend | <0.001 | | <0.001 | | <0.001 | | 0.002 | |

[a]Model I: Adjusted for age, sex, and profession.

[b]Model II: Adjusted for age, sex, profession, and comorbidity.

[c]Model III: Adjusted for all covariates.

Abbreviations: HR, hazard ratio; CI, confidence interval.

## Discussion

In this retrospective study, we found that serum ferritin levels in patients with SFTS are an important inflammatory marker that is linearly associated with in-hospital mortality. Specifically, higher serum ferritin levels were correlated with increased mortality risk in patients with SFTS, with levels ≥10,000 ng/ml associated with a notably higher mortality risk.

We observed that serum ferritin levels in patients who died were significantly higher than those in patients who survived, which is consistent with the findings of Chen et al. [13]. Using a Cox proportional risk model, we determined that serum ferritin levels were significantly associated with in-hospital mortality, regardless of adjustments for other covariates. This further confirms the reliability of serum ferritin level as a prognostic indicator for patients with SFTS. Notably, a linear relationship between ferritin levels and in-hospital mortality was evident across all four Cox proportional risk models, indicating a consistent trend of increased mortality risk with increasing ferritin levels.

Serum ferritin levels may reflect a balance between beneficial immune responses and a harmful inflammatory overreaction in patients with COVID-19 [14–15], Papamanoli et al. [16] posited that serum ferritin levels up to 1,300 ng/mL potentially indicating a healthy immune response to acute infections, such as COVID-19. However, Kooistra et al. [17] reported that serum ferritin levels >1300 ng/ml may indicate a dysregulated, harmful inflammatory response. Likewise, Lalueza et al. [8] reported that patients with influenza virus infection with serum ferritin levels >609 ng/ml tended to have a poorer prognosis. In this study, we found that patients with SFTS with serum ferritin levels >10,000 ng/ml had a significantly elevated mortality risk, underscoring the potential of serum ferritin levels as an effective predictor of in-hospital mortality

in patients with SFTS. In addition, we further validated the stability and reliability of serum ferritin levels in the prognostic assessment of patients using PSM and trend tests.

The mechanism by which SFTSV infection leads to an elevated serum ferritin level remains unclear. It might be related to viral infection that could disrupt iron metabolism within the body, causing iron release and redistribution and thereby further increasing ferritin production. It could also be associated with an excessive immune response triggered by SFTSV infection. Viral infection activates immune cells and promotes the release of inflammatory factors [18]. These inflammatory factors, in turn, stimulate macrophages and other immune cells to secrete ferritin, raising serum ferritin levels [19]. Additionally, HLH is another disorder associated with significantly elevated serum ferritin levels. The disease can be secondary to viral infections (particularly Epstein-Barr virus and cytomegalovirus) and is clinically characterized by pancytopenia, hypertriglyceridemia, methemoglobinemia, and multiple organ failure, often with fatal outcomes when serum ferritin levels typically exceed 10,000 μg/L [20]. Ferritin binds to serum IL-2 receptor-α and other indicators to offer diagnostic support [21]. A study of an animal model of SFTS virus infection showed that SFTS virus-induced thrombocytopenia is caused by activated splenic macrophages clearing circulating virus-bound platelets. These findings suggest that SFTS might be one of the infectious causes of HLH, and based on animal studies, activated macrophages in the spleen may exhibit hemophagocytic activity, resulting in hyperferritinemia [6]. Jia et al. demonstrated that ferritin is directly involved in the formation of cytokine storms, thus triggering systemic inflammatory response and leading to multiple organ dysfunction [22–23]. In our study, weobserved a markedly increased mortality rate among patients with ferritin levels exceeding 10,000 ug/L, which may be related to both secondary HLH in patients and the associated sepsis induced by SFTSV and accelerated organ failure [24]. However, Kimura et al. found that the intracellular pathogen Mycobacterium tuberculosis induces a high level of ferritin secretion in human monocytes (THP-1 cells) through unconventional secretional autophagy activation compared to LPS-treated THP1 cells. These findings indicated that serum ferritin can increase due to intracellular infection itself, independent of the effects of cytophagy [25]. Furthermore, serum ferritin may act as an anti-inflammatory mediator by binding to high molecular weight potassium kininogen and inhibiting its division, leading to reduced bradykinin release and a decreased total inflammatory response [26]. Therefore, whether serum ferritin is a pro-inflammatory or anti-inflammatory marker remains unclear, and more studies are required to determine the exact source and mechanism of hyperferritinemia in SFTS patients in the future.

Our study has some limitations. First, this was a single-center retrospective study; therefore, it may have been subject to selection bias. Second, although we used PSM to reduce bias, residual confounding factors may remain. Third, we did not assess the impact of certain treatments, such as hormone use, on outcomes. Finally, we did not assess dynamic changes in serum ferritin levels over time. Therefore, prospective randomized controlled trials are needed to confirm the effect of serum ferritin levels on the prognosis of patients with SFTS.

## Conclusion

Serum ferritin levels can be used as a reliable indicator of the prognosis of patients with SFTS. Monitoring serum ferritin levels may help clinicians assess the mortality risk and formulate more targeted treatment plans. However, further prospective studies with larger sample sizes are required to validate these findings.

## Supporting information

**S1 Table. Crude model.**
(DOCX)

**S2 Table. Model I.**
(DOCX)

**S3 Table. Model II.**
(DOCX)

**S4 Table. Model III.**
(DOCX)

**S1 Fig. Covariate balance plot.**
(TIF)

**S1 Data. Clinical and laboratory data of patients with SFTS.**
(XLSX)

**S2 Data. Data such as specificity, sensitivity and AUC under different cut-off values.**
(XLSX)

## Acknowledgments

We would like to thank all our colleagues in the Department of Infectious Diseases and Second Department of Critical Care Medicine of the Second Affiliated Hospital of Anhui Medical University and Professor Liu Yu of Anhui University for their support.

## Author contributions

**Conceptualization:** Wenyan Xiao, Min Yang.

**Data curation:** Wenyan Xiao, Liangliang Zhang, Juanjuan Hu.

**Formal analysis:** Liangliang Zhang, Yang Zhang, Jin Zhang.

**Funding acquisition:** Wenyan Xiao, Min Yang.

**Investigation:** Wenyan Xiao, Liangliang Zhang, Tianfeng Hua.

**Methodology:** Wenyan Xiao, Yang Zhang.

**Project administration:** Wenyan Xiao, Liangliang Zhang, Juanjuan Hu, Tianfeng Hua.

**Resources:** Juanjuan Hu, Jin Zhang, Tianfeng Hua.

**Software:** Liangliang Zhang, Yang Zhang.

**Supervision:** Tianfeng Hua, Min Yang.

**Writing – original draft:** Wenyan Xiao, Yang Zhang.

**Writing – review & editing:** Tianfeng Hua, Min Yang.

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
