## [Decision Letter · Decision Letter 0]

18 Feb 2025

PNTD-D-24-01734

Association between serum ferritin and mortality in patients with severe fever with thrombocytopenia syndrome�a retrospective cohort study

Dear Dr. Yang,

Thank you for submitting your manuscript to PLOS Neglected Tropical Diseases. After careful consideration, we feel that it has merit but does not fully meet PLOS Neglected Tropical Diseases's publication criteria as it currently stands. Therefore, we invite you to submit a revised version of the manuscript that addresses the points raised during the review process.

Please submit your revised manuscript within 60 days Apr 19 2025 11:59PM. If you will need more time than this to complete your revisions, please reply to this message or contact the journal office at plosntds@plos.org. Please include the following items when submitting your revised manuscript:

We look forward to receiving your revised manuscript.

Kind regards,

Gregory Gromowski

Academic Editor

Abdallah Samy

Section Editor

Shaden Kamhawi

co-Editor-in-Chief

Paul Brindley

co-Editor-in-Chief

**Journal Requirements:**

1) Tables should not be uploaded as individual files. Please remove these files and include the Tables in your manuscript file as editable, cell-based objects. For more information about how to format tables, see our guidelines:

https://journals.plos.org/plosntds/s/tables

2) We have noticed that you have uploaded Supporting Information files, but you have not included a list of legends. Please add a full list of legends for your Supporting Information files after the references list.

3) In the online submission form, you indicated that "The dataset used in this study is available from the corresponding author upon reasonable request". All PLOS journals now require all data underlying the findings described in their manuscript to be freely available to other researchers, either

- In a public repository

- Within the manuscript itself

- Uploaded as supplementary information.

4) Please amend your detailed Financial Disclosure statement. This is published with the article. It must therefore be completed in full sentences and contain the exact wording you wish to be published. Please ensure that the funders and grant numbers match between the Financial Disclosure field and the Funding Information tab in your submission form. Note that the funders must be provided in the same order in both places as well.

**Reviewers' Comments:**

Reviewer's Responses to Questions

**Key Review Criteria Required for Acceptance?**

**Methods** :

-Are the objectives of the study clearly articulated with a clear testable hypothesis stated?

-Is the study design appropriate to address the stated objectives?

-Is the population clearly described and appropriate for the hypothesis being tested?

-Is the sample size sufficient to ensure adequate power to address the hypothesis being tested?

-Were correct statistical analysis used to support conclusions?

-Are there concerns about ethical or regulatory requirements being met?

Reviewer #1: T

Reviewer #2: Wenyan Xiao et al retrospective analysis using data from patients diagnosed with SFTS at the Second Affiliated Hospital of Anhui Medical University,and found the association between serum ferritin levels and in-hospital mortality in patients with SFTS, a study with a relatively large sample, also had some important findings.

Reviewer #3: (No Response)

**Results** :

-Does the analysis presented match the analysis plan?

-Are the results clearly and completely presented?

-Are the figures (Tables, Images) of sufficient quality for clarity?

Reviewer #1: (No Response)

Reviewer #2: 1) Single-center, retrospective study.

2) The changes of serum ferritin levels were not dynamically displayed and analyzed.

Reviewer #3: After PSM, SMD should be reported.

**Conclusions** :

-Are the conclusions supported by the data presented?

-Are the limitations of analysis clearly described?

-Do the authors discuss how these data can be helpful to advance our understanding of the topic under study?

-Is public health relevance addressed?

Reviewer #1: (No Response)

Reviewer #2: 3) The stage changes of acute viral infection and the course of the disease significantly affect the results of laboratory tests, that is, in the critical stage of SFTS infection, especially when accompanied by multiple organ damage such as pancreatitis, encephalitis, etc., the inflammatory markers and serum ferritin levels of patients will be significantly increased.

4) Therefore, the level of serum ferritin is uncertain in the course of disease, but it will obviously increase in the case of multiple organ inflammation. At this time, the predictive role of serum ferritin is not very important.

Reviewer #3: Further explanation is needed regarding why serum ferritin is associated with increased mortality. Is it due to sepsis (DOI: 10.3389/fnut.2021.747547) or cytokine storm (DOI: 10.1038/s41467-022-34560-7; 10.1038/s41584-022-00899-w)?

**Editorial and Data Presentation Modifications?**

Reviewer #1: The paper reported an association between serum ferritin and mortality in patients with severe fever with thrombocytopenia syndrome (SFTS). It was found that serum ferritin levels are linearly associated with the mortality risk in SFTS patients. Specifically, when serum ferritin levels exceed 10,000 ng/ml, the mortality rate significantly increases. Thus, serum ferritin levels may serve as a valuable prognostic biomarker for assessing the mortality risk in SFTS patients. The study is well - designed, and its results are highly useful for predicting the prognosis of SFTS.

1. Results: Regarding the results section, particularly the "receiver operating characteristic (ROC) curve of serum ferritin for predicting in - hospital mortality in patients with SFTS", the authors are recommended to offer more detailed information. This could include aspects such as the area under the curve (AUC) value, the optimal cut - off point determined from the ROC analysis, and the sensitivity and specificity values at different thresholds. Such additional details would enhance the comprehensiveness and interpretability of the results.

2. Serum ferritin is generally regarded as having both pro - inflammatory and anti - inflammatory properties. Its function is intricate and context - dependent. As a result, it cannot be simply classified as a strictly pro - inflammatory or anti - inflammatory marker. In different physiological and pathological conditions, serum ferritin can exhibit diverse effects on the inflammatory response. For example, in certain acute - phase responses, it may act as a pro - inflammatory factor by being upregulated in response to inflammatory cytokines. However, in maintaining iron homeostasis, it can play an anti - inflammatory role by regulating the availability of iron, which is crucial for immune cell function and the prevention of oxidative stress - related inflammation.

3. Throughout the text, some minor adjustments have been made to enhance the fluency and clarity of the English.

Reviewer #2: (No Response)

Reviewer #3: (No Response)

**Summary and General Comments** :

Reviewer #1: (No Response)

Reviewer #2: (No Response)

Reviewer #3: (No Response)

PLOS authors have the option to publish the peer review history of their article (what does this mean? ). If published, this will include your full peer review and any attached files.

**Do you want your identity to be public for this peer review?** For information about this choice, including consent withdrawal, please see our Privacy Policy .

Reviewer #1: **Yes: ** Quan Liu

Reviewer #2: No

Reviewer #3: No

**Figure resubmission:**
---

## [Decision Letter · Decision Letter 1]

4 May 2025

Dear Dr. Yang,

We are pleased to inform you that your manuscript 'Association between serum ferritin and mortality in patients with severe fever with thrombocytopenia syndrome�a retrospective cohort study' has been provisionally accepted for publication in PLOS Neglected Tropical Diseases.

Best regards,

Gregory Gromowski

Academic Editor

Abdallah Samy

Section Editor

Shaden Kamhawi

co-Editor-in-Chief

Paul Brindley

co-Editor-in-Chief

---

## [Editor Report · Acceptance letter]

Dear Dr. Yang,

We are delighted to inform you that your manuscript, "Association between serum ferritin and mortality in patients with severe fever with thrombocytopenia syndrome�a retrospective cohort study," has been formally accepted for publication in PLOS Neglected Tropical Diseases.

Best regards,

Shaden Kamhawi

co-Editor-in-Chief

Paul Brindley

co-Editor-in-Chief
